# Reactive Chlorine Species Reversibly Inhibit DnaB Protein Splicing in Mycobacteria

Christopher W. Lennon,[a] Daniel Wahl,[a] J. R. Goetz,[a] Joel Weinberger, II[a]

aDepartment of Biological Sciences, Murray State University, Murray, Kentucky, USA

**ABSTRACT** Intervening proteins, or inteins, are mobile genetic elements that are translated within host polypeptides and removed at the protein level by splicing. In protein splicing, a self-mediated reaction removes the intein, leaving a peptide bond in place. While protein splicing can proceed in the absence of external cofactors, several examples of conditional protein splicing (CPS) have emerged. In CPS, the rate and accuracy of splicing are highly dependent on environmental conditions. Because the activity of the intein-containing host protein is compromised prior to splicing and inteins are highly abundant in the microbial world, CPS represents an emerging form of posttranslational regulation that is potentially widespread in microbes. Reactive chlorine species (RCS) are highly potent oxidants encountered by bacteria in a variety of natural environments, including within cells of the mammalian innate immune system. Here, we demonstrate that two naturally occurring RCS, namely, hypochlorous acid (the active compound in bleach) and *N*-chlorotaurine, can reversibly block splicing of DnaB inteins from *Mycobacterium leprae* and *Mycobacterium smegmatis in vitro*. Further, using a reporter that monitors DnaB intein activity within *M. smegmatis*, we show that DnaB protein splicing is inhibited by RCS in the native host. DnaB, an essential replicative helicase, is the most common intein-housing protein in bacteria. These results add to the growing list of environmental conditions that are relevant to the survival of the intein-containing host and influence protein splicing, as well as suggesting a novel mycobacterial response to RCS. We propose a model in which DnaB splicing, and therefore replication, is paused when these mycobacteria encounter RCS.

**IMPORTANCE** Inteins are both widespread and abundant in microbes, including within several bacterial and fungal pathogens. Inteins are domains translated within host proteins and removed at the protein level by splicing. Traditionally considered molecular parasites, some inteins have emerged in recent years as adaptive posttranslational regulatory elements. Several studies have demonstrated CPS, in which the rate and accuracy of protein splicing, and thus host protein functions, are responsive to environmental conditions relevant to the intein-containing organism. In this work, we demonstrate that two naturally occurring RCS, including the active compound in household bleach, reversibly inhibit protein splicing of *Mycobacterium leprae* and *Mycobacterium smegmatis* DnaB inteins. In addition to describing a new physiologically relevant condition that can temporarily inhibit protein splicing, this study suggests a novel stress response in *Mycobacterium*, a bacterial genus of tremendous importance to humans.

**KEYWORDS** intein, conditional protein splicing, reactive chlorine species, bleach, chloramines, mycobacteria, DnaB helicase

Inteins are intervening proteins that excise themselves from precursor polypeptides through protein splicing. In this process, the intein is removed and the flanking N and C exteins are joined together with a peptide bond to yield a mature host protein

Address correspondence to Christopher W. Lennon, clennon1@murraystate.edu.

(1–5). In the nearly three decades since the discovery of inteins, thousands have been identified across the genomes of archaea, bacteria, unicellular eukaryotes, phages, and viruses. Inteins are particularly abundant in prokaryotes; they are found in almost one-quarter of bacterial genomes and one-half of archaeal genomes (6).

The protein splicing reaction is mediated by conserved residues most often found near the intein-extein junctions. In class 1 protein splicing (7), the first residue of the intein (either cysteine or serine) makes a nucleophilic attack on the preceding peptide bond, leading to an ester/thioester linkage. Next, the first residue of the C extein (either cysteine, serine, or threonine) makes another nucleophilic attack on the ester/thioester formed in step 1, leading to formation of the branched intermediate. Following this, the terminal asparagine of the intein cyclizes to release the intein. Finally, the ester/thioester bond connecting the N and C exteins rearranges to form a peptide bond. Two alternative splicing mechanisms exist (class 2 and class 3), with an overall conservation of intein fold and chemistry but important differences in the catalytic steps. For class 3 inteins, step 1 is carried out by an internal nucleophile (Fig. 1A), with steps 2 to 4 occurring in a manner similar to that for class 1 inteins (7). Off-pathway reactions can also occur, cleaving the N or C extein from the intein prior to ligation.

The ability of inteins to rearrange or cleave peptide bonds in a controlled manner has been exploited in numerous biotechnological applications, including protein purification, segmental isotope labeling, formation of cyclic peptides, incorporation of non-natural modules into proteins, fabrication of protein arrays, sensor development, imaging, and regulation of protein function *in vivo* (8, 9). Additionally, new intein-based technologies frequently emerge (10–13). Given the useful chemistry performed by inteins, as well as the ability to leave no trace in the final product, the potential of these elements in protein engineering is tremendous.

Inteins themselves often house an autonomous homing endonuclease (HEN) domain that is not directly involved in protein splicing but rather is involved in horizontal transfer of the intein-coding DNA. Therefore, inteins have been traditionally viewed as selfish genetic elements, the invasiveness of which helps remodel genomes (14, 15). However, several intriguing aspects of intein distribution compel us to rethink the role of these elements in nature. Inteins cluster in particular functional classes of proteins, with over 70% being found in ATP-binding proteins and over 60% being localized in proteins involved in DNA replication, recombination, and repair (6). Remarkably, inteins have even been independently acquired by evolutionarily distinct bacterial and archaeal proteins of equivalent function (6). Further, inteins are typically found within genes essential for host organism survival, with their presence disrupting function. Therefore, successful protein splicing is required for the survival of a significant fraction of all life.

In the past decade, a growing body of evidence has demonstrated that some inteins undergo conditional protein splicing (CPS), whereby splicing rate and formation of off-pathway products are intimately tied to environmental conditions. We and others have challenged the model of pure selfishness and argue that many inteins, particularly inteins that have lost their HEN domain, have transitioned to a form of "altruism." Several studies have provided compelling examples of CPS in which splicing is highly dependent on environmental conditions. These conditions, which are often crucial for the survival of the host organisms and/or relevant to the function of the intein-containing protein, include pH, divalent cations, redox, reactive oxygen species (ROS)/reactive nitrogen species (RNS), salt, temperature, and even DNA damage (single-stranded DNA [ssDNA]) (16–29). Therefore, some inteins can seemingly modulate protein splicing in response to stress, regulating intein-housing protein activity.

Reactive chlorine species (RCS) modify several biomolecules, including amino acid side chains, nucleotides, and lipids (30, 31). Sulfur-containing amino acid side chains, such as cysteine, are particularly susceptible to RCS oxidation and chlorination, reacting ~100-fold faster than other biomolecules (30, 31). Many bacteria encounter RCS in nature, such as hypochlorous acid (HOCl), the active compound in bleach, and have

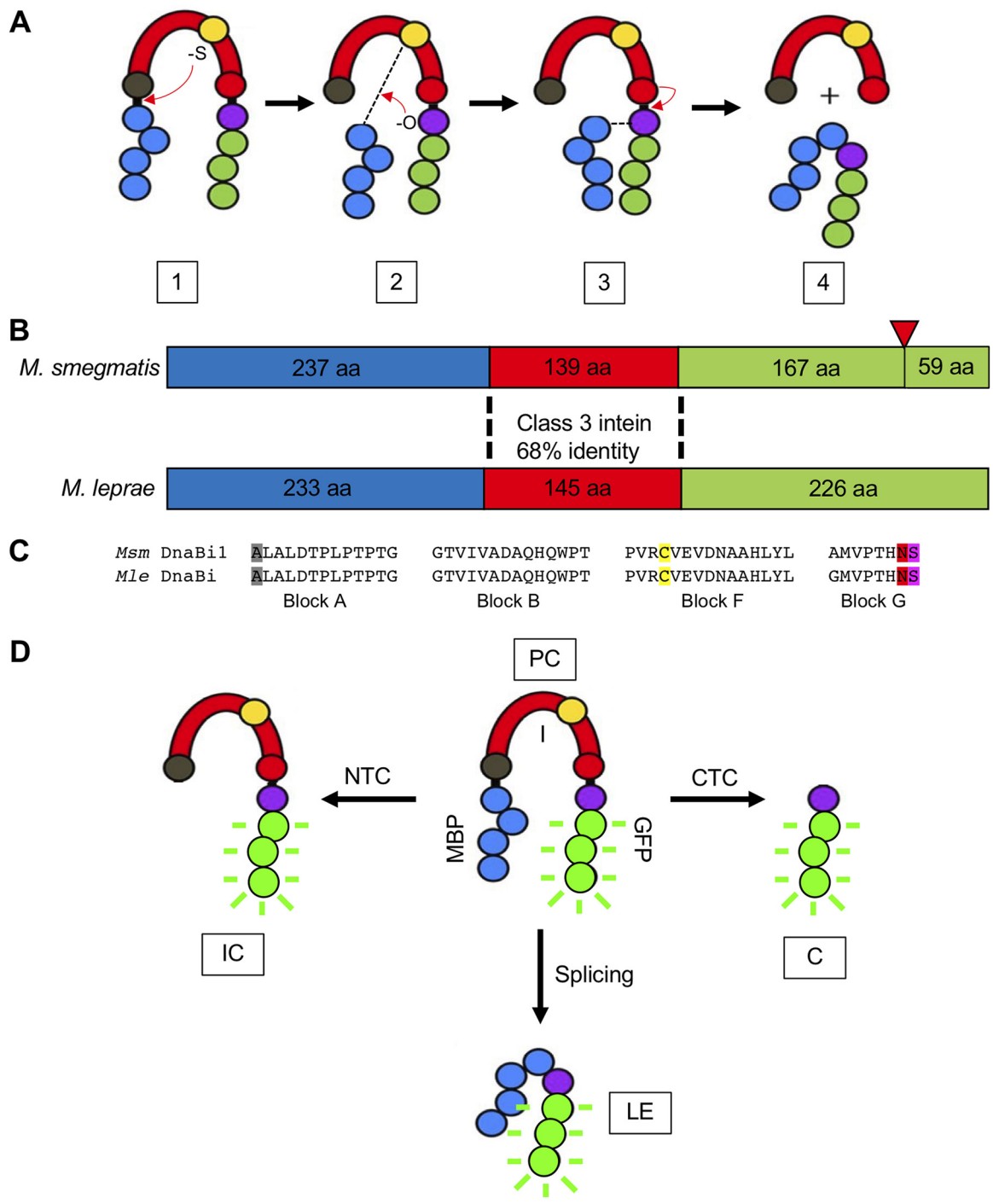

**FIG 1** *M. smegmatis* DnaBi1 and *M. leprae* DnaBi. (A) Mechanism of class 3 protein splicing. Steps 1 to 4 are described in the text. Blue, N extein; red, intein; green, C extein; gray ball, first intein residue; yellow ball, intein initiating nucleophile; red ball, intein terminal asparagine; purple ball, +1 residue; dashed line, thioester linkage. (B) Comparison of *M. smegmatis* DnaBi1 and *M. leprae* DnaBi intein insertion positions in DnaB. aa, amino acids. Blue, N extein; red, intein; green, C extein; red inverted triangle, position of second intein within *M. smegmatis* that is absent in *M. leprae*. (C) Comparison of splicing blocks between *M. smegmatis* DnaBi1 and *M. leprae* DnaBi. Gray, first intein residue; yellow, intein initiating nucleophile; red, intein terminal asparagine; purple, +1 nucleophile. (D) MIG intein activity reporter. Species visible through GFP fluorescence are shown, including unspliced precursor (PC), ligated exteins (LE) from splicing, intein-C extein (IC) from N-terminal cleavage (NTC), and C extein (C) from C-terminal cleavage (CTC).

evolved specific response pathways (30, 31). For example, cells of the mammalian innate immune system produce RCS to kill foreign bacteria (30, 31). Within neutrophils, myeloperoxidase (MPO) produces millimolar quantities of HOCl from hydrogen peroxide ($H_2O_2$) and chloride (31). Chloramines, which are produced upon HOCl reaction

with amines, represent another highly reactive, naturally occurring RCS with potent antimicrobial properties (30). For example, *N*-chlorotaurine (NCT), a derivative of the amino acid taurine, is a relatively long-lived RCS that is naturally produced by granulocytes and monocytes (32). Interestingly, while chloramines are less reactive than HOCl, they are more specific for cysteine oxidation (30).

Here, we demonstrate that two physiologically relevant RCS, namely, NCT and HOCl, can inhibit protein splicing of DnaB inteins from *Mycobacterium leprae* and *Mycobacterium smegmatis*. Additionally, we find that this inhibition is fully reversible in the presence of reducing agent. Finally, using an *in vivo* reporter that directly links intein activity to cell survival in *M. smegmatis*, we show that DnaB intein splicing can be inhibited within the natural context of the intein. We propose a model whereby RCS can act to reversibly pause protein splicing stress. We argue that the ability of some inteins to detect stress and temporarily pause critical cellular processes, such as DNA replication, may help the intein-containing organism survive and persist under harsh conditions.

## RESULTS

The helicase DnaB is the most abundant intein-containing protein in bacteria and is strictly required for DNA replication (6). *M. smegmatis* and *M. leprae* each house a class 3 intein within DnaB, referred to as *M. smegmatis* DnaBi1 and *M. leprae* DnaBi, respectively. *M. smegmatis* DnaB also houses a second, full-length intein (Fig. 1B) that splices fully upon expression (25) and does not appear to be subject to CPS. *M. smegmatis* DnaBi1 and *M. leprae* DnaBi both localize within the P-loop of the DnaB ATPase domain (Fig. 1B), with the +1 serine of the C extein acting to help coordinate $Mg^{2+}$-ATP following splicing (25). *M. smegmatis* DnaBi1 and *M. leprae* DnaBi are both mini-inteins. Because mini-inteins lack a HEN domain and are no longer capable of endonuclease-mediated mobility, these represent especially intriguing candidates for CPS. *M. smegmatis* DnaBi1 and *M. leprae* DnaBi residues are 68% identical overall (Fig. 1B) and splicing blocks are highly conserved, varying only in the first position of block G (Fig. 1C). Further, both *M. smegmatis* DnaBi1 and *M. leprae* DnaBi use cysteine as the initiating nucleophile to begin the splicing process. Because these are class 3 inteins, this catalytic cysteine is internal (C-118 for *M. smegmatis* DnaBi1 and C-124 for *M. leprae* DnaBi), rather than being the first residue of the intein.

We tested the hypothesis that the activity of these inteins could be blocked by RCS. To measure the potential inhibition of protein splicing, we utilized an established splicing reporter referred to as MIG (MBP-intein-GFP). In the MIG reporter, the intein (surrounded by 10 native extein residues) is flanked by the nonnative exteins maltose-binding protein (MBP) (N extein) and superfolder green fluorescent protein (GFP) (C extein). This reporter can be used to visualize precursor, ligated exteins, and products resulting from off-pathway cleavage (intein-C extein from N terminal cleavage and C extein from C-terminal cleavage) (Fig. 1D). GFP-containing products (i.e., C extein) can be detected using seminative PAGE with fluorescent products measured in the gel (33). Additionally, this reporter has a built-in control for generalized protein misfolding, because the structure of GFP must remain intact in order to maintain the fluorescence signal.

Following expression in the MIG reporter, as observed previously, most *M. smegmatis* DnaBi1 was in the unspliced precursor (Fig. 2A, C, and D). We next treated *M. smegmatis* DnaBi1 with two naturally occurring RCS, namely, the chloramine NCT and HOCl. For both of these RCS, we observed substantial inhibition of splicing (Fig. 2A and D). Interestingly, we observed a second precursor band form (Fig. 2C), a pattern previously observed when *M. smegmatis* DnaBi1 was treated with $H_2O_2$ and formed an intramolecular disulfide. We also observed a faint band consistent with an intermolecular disulfide bond between precursors (Fig. 3A). Critical to the model that inteins can serve as "pause buttons" to temporarily block splicing under stress (29), we next asked whether this observed inhibition was reversible. Dithiothreitol (DTT), a reducing agent, is

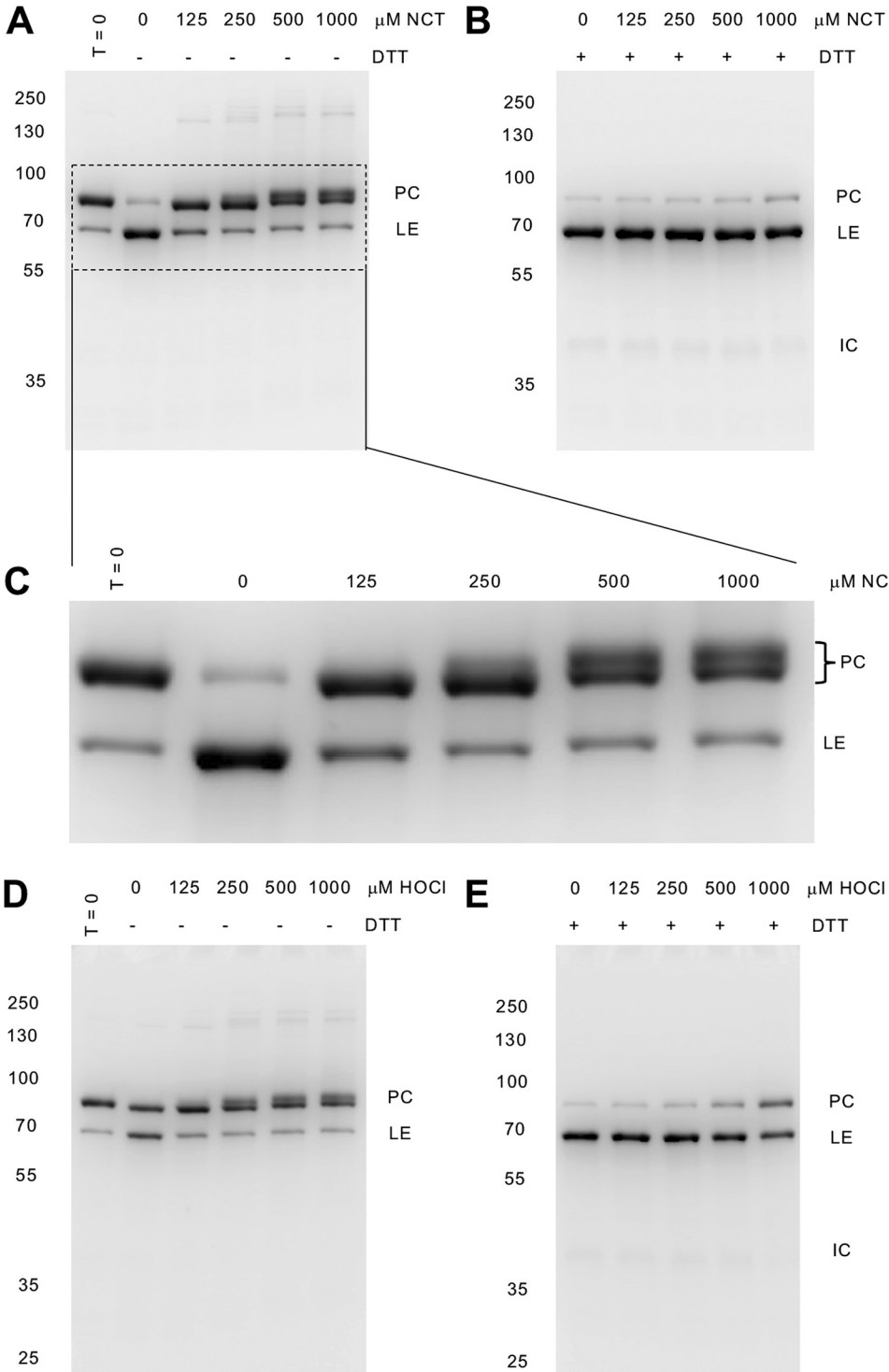

**FIG 2** *M. smegmatis* DnaBi1 splicing is reversibly inhibited by RCS. (A) NCT treatment of *M. smegmatis* DnaBi1. The concentrations of NCT are indicated. (B) Excess (50 mM) DTT (reducing agent) reverses the inhibition by NCT and permits splicing. (C) Enlargement of a gel section from panel A. (D) HOCl treatment of *M. smegmatis* DnaBi1 at the concentrations indicated. (E) Reversal of *M. smegmatis* DnaBi1 splicing inhibition by HOCl in the presence of excess DTT (50 mM). Each gel is representative of at least three independent experiments. Molecular weights are indicated to the left of the gels, and GFP-containing species are indicated to the right of the gels. PC, unspliced precursor; LE, ligated exteins; IC, intein-C extein.

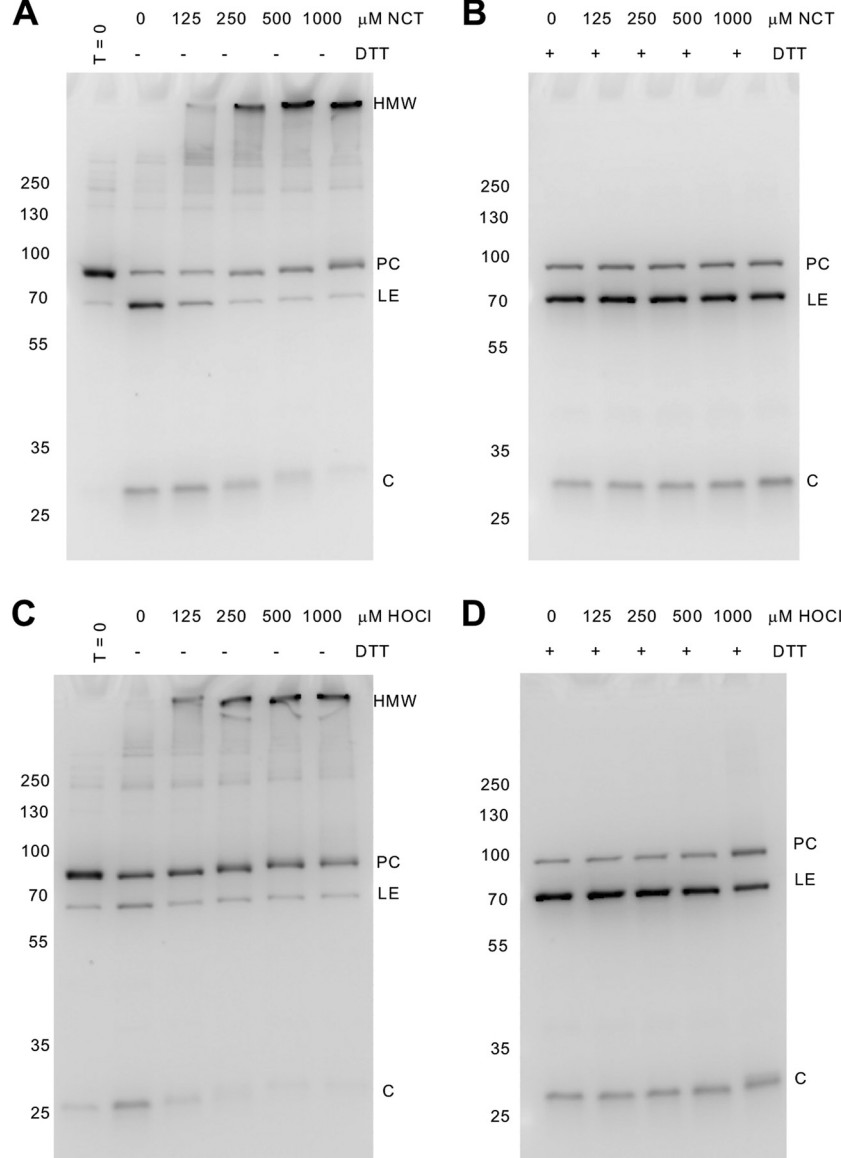

**FIG 3** Reversible inhibition of *M. leprae* DnaBi activity by RCS. (A) NCT treatment inhibits *M. leprae* DnaBi splicing. The concentrations of NCT are indicated. (B) DTT in excess (50 mM) reverses the inhibition by NCT. (C) *M. leprae* DnaBi activity is blocked by HOCl at the indicated concentrations. (D) *M. leprae* DnaBi activity is restored following excess DTT (50 mM) treatment. Each gel is representative of at least three independent experiments. Molecular weights are indicated to the left of the gels, and GFP-containing species are indicated to the right of the gels. PC, unspliced precursor; LE, ligated exteins; C, C extein.

capable of reversing some RCS modifications. To test this, we first incubated lysate with either HOCl or NCT, followed by an excess of the reducing agent DTT, which is expected to reverse oxidative RCS modifications on cysteines (30). Consistent with reversible intein modification, we found that the inhibition of splicing by RCS was fully reversible following treatment with DTT (Fig. 2B and E).

We next asked whether *M. leprae* DnaBi activity was also inhibited by RCS. As with *M. smegmatis* DnaBi1, substantial levels of unspliced precursor remained following expression in the MIG reporter (Fig. 3A and C). Of note, we observed significant C-terminal cleavage, as indicated by the free C extein, for *M. leprae* DnaBi; this can also be used as a measure of intein activity. Following treatment with NCT and HOCl, *M. leprae* DnaBi splicing was also fully inhibited (Fig. 3A and C). Unlike *M. smegmatis* DnaBi1, *M.*

*leprae* DnaBi appeared to form a high-molecular-weight (HMW) product (Fig. 3A and C). This product is soluble, and GFP in the MIG reporter remains folded. Importantly, as with *M. smegmatis* DnaBi1, *M. leprae* DnaBi activity was restored upon incubation with DTT (Fig. 3B and D). This HMW band likely represents several precursors forming intermolecular disulfide bonds. We also observed that RCS changed the migration of the C extein (Fig. 3A and C), likely through modification of the C extein that is reversible with DTT (Fig. 3B and D).

Examination of CPS *in vivo* has traditionally proved challenging due to the lack of inteins in model organisms. Previously, we developed a splicing reporter that can be used directly within *M. smegmatis* cells to monitor *M. smegmatis* DnaBi1 activity. In our kanamycin intein splicing reporter (KISR), *M. smegmatis* DnaBi1 is inserted between residue 153 and residue 154 of the aminoglycoside phosphotransferase KanR (NCBI reference sequence WP_000018329.1) (Fig. 4A). *M. smegmatis* DnaBi1 splicing is strictly required for resistance, as mutation of the initiating nucleophile C-118 renders *M. smegmatis* cells sensitive to kanamycin (27, 34) (Fig. 4B). Oxidative stress ($H_2O_2$) and $Zn^{2+}$ were both shown to reduce *M. smegmatis* survival in an intein-dependent manner using this reporter, and the results were confirmed *in vitro* (25, 27).

To test whether RCS could also inhibit *M. smegmatis* DnaBi1 activity *in vivo*, equal numbers of *M. smegmatis* cells expressing either KanR or KISR with wild-type (WT) *M. smegmatis* DnaBi1 (KISR-WT) were spread on plates with kanamycin alone or kanamycin with NCT. In the presence of kanamycin alone, *M. smegmatis* cells expressing KanR or KISR-WT displayed equal survival rates (Fig. 4C). As expected, NCT (0.2 mM) in addition to kanamycin led to reductions in survival rates for both KanR- and KISR-WT-expressing strains (Fig. 4C). In the presence of NCT and kanamycin, however, the survival rate for *M. smegmatis* with KISR-WT was substantially reduced, compared to the survival rate for *M. smegmatis* with KanR (Fig. 4C). Based on CFU counts from three independent experiments, this corresponded to an average 91-fold reduction in the survival rate for the KISR-WT-expressing strain. This is similar in magnitude to the effects observed previously with oxidative stress and $Zn^{2+}$ (25, 27). This observed reduction was variable, with a standard deviation of 46% for the fold reduction, likely due to the volatility of NCT. The overall trend was constant, however, with the KISR-WT-expressing strain always being more sensitive with NCT and kanamycin.

## DISCUSSION

DnaB is the most common intein-housing protein in bacteria (6). Previous work demonstrated that mycobacterial DnaB splicing is subject to CPS in the presence of $H_2O_2$ and $Zn^{2+}$, with reversible inhibition of *M. smegmatis* DnaBi1 occurring in both cases (25, 27). In the case of $H_2O_2$ stress, *M. smegmatis* DnaBi1 forms a reversible, intramolecular disulfide bond that includes the initiating nucleophile C-118 (25). For $Zn^{2+}$, this metal binds directly to *M. smegmatis* DnaBi1, with C-118 forming part of the binding pocket (27). Given the finding that RCS are highly reactive with cysteine side chains (30), the tendency of inteins to utilize cysteine for splicing, and the abundance of inteins in the microbial world, our findings in this work demonstrate the potential for RCS sensing by inteins in nature. Importantly, inhibition by RCS of these two DnaB inteins is reversible in the presence of reducing agent (Fig. 2 and 3). Further, our results using our *in vivo* KISR demonstrate that splicing of *M. smegmatis* DnaBi1 can be blocked in the mycobacterial cellular environment (Fig. 4).

Discovering new physiologically relevant environmental conditions that influence protein splicing is crucial to understanding the biological roles for inteins in nature. Further, inteins represent attractive drug targets because they are found within essential genes of human pathogens and are absent in metazoans (6). Toward this end, the anticancer drug cisplatin was shown to inhibit the growth of *Mycobacterium tuberculosis* in an intein-specific manner (35) and was shown by crystallography to bind directly to catalytic residues of the RecA intein (36). Recent evidence suggests that cisplatin also targets an intein within the essential splicesosomal Prp8 protein of the pathogen *Cryptococcus neoformans* (37, 38).

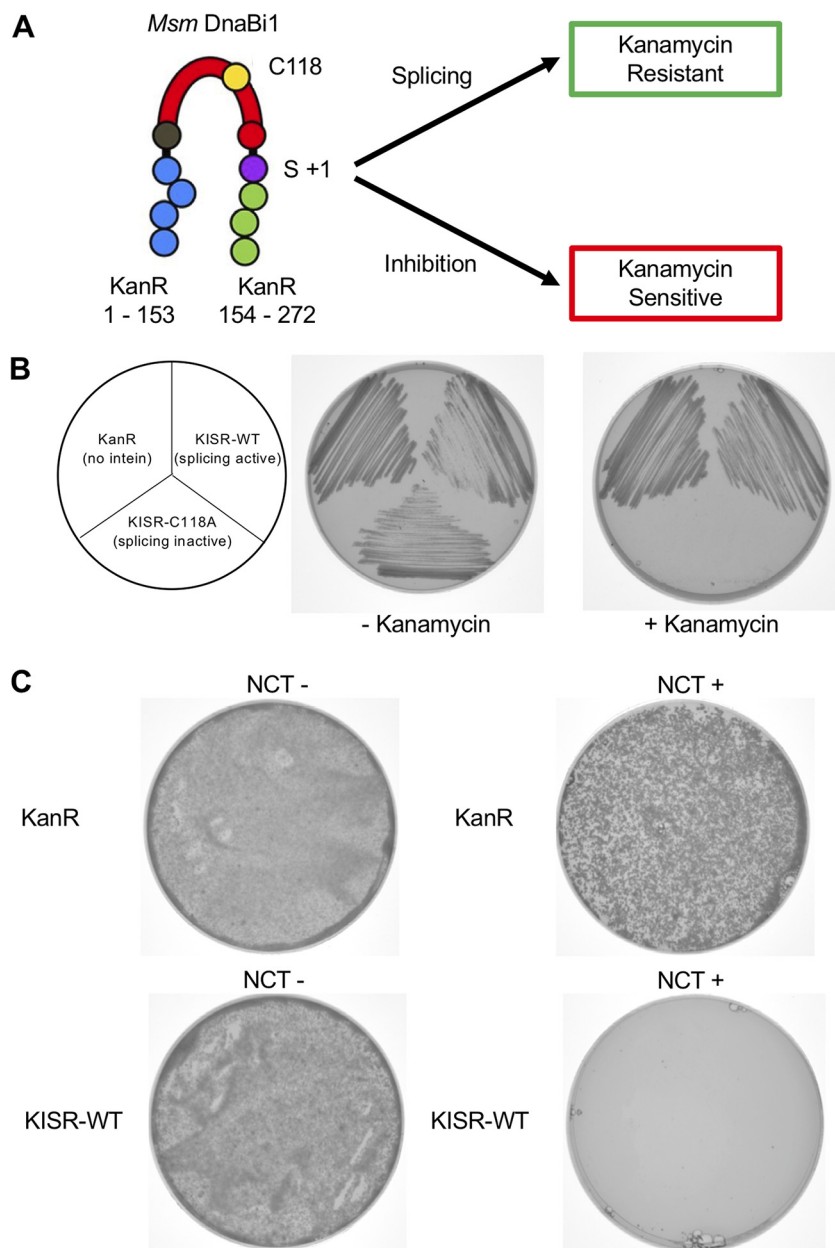

**FIG 4** RCS inhibit *M. smegmatis* DnaBi1 splicing-dependent survival in *M. smegmatis* cells. (A) KISR, in which *M. smegmatis* DnaBi1 interrupts the KanR protein and splicing is strictly required for resistance to kanamycin. (B) KanR and KISR-WT provide resistance to kanamycin. KISR-C118A, in which the initiating nucleophile C-118 of *M. smegmatis* DnaBi1 has been mutated, is not resistant to kanamycin. (C) NCT (0.2 mM) dramatically inhibits survival rates of *M. smegmatis* cells expressing KISR. Equal amounts of KanR-expressing (no intein) or KISR-expressing *M. smegmatis* cells were plated in the presence of kanamycin and in the presence or absence of NCT. While KanR-expressing and KISR-expressing *M. smegmatis* cells survive to the same extent in the absence of NCT (left), KISR-expressing cell survival rates are substantially decreased, compared to KanR-expressing cells, in the presence of NCT (right).

Thus, understanding the conditions that control the splicing of inteins from pathogens such as *M. leprae* not only yields insights into their responses to stress but also may inform the design of intein-specific inhibitors that selectively kill them.

Bacteria encounter RCS in several natural contexts, and numerous defense mechanisms have been described (30, 31). One such environment is in cells of the mammalian innate immune system. Within the phagolysosome, MPO produces HOCl from $H_2O_2$ and chloride (31). MPO, which can represent 5% of the total protein in neutrophils (30), can

produce levels of HOCl that reach 25 to 50 mM (39). *M. tuberculosis* and *M. leprae*, the agents responsible for tuberculosis and leprosy, respectively, can survive within the phagolysosome (40). The observed concentrations of RCS that inhibit *M. smegmatis* and *M. leprae* DnaB splicing are substantially (100 to 200-fold) below the levels of RCS produced within neutrophils, far below the physiological concentrations produced.

RCS are highly reactive with cysteines and can lead to several types of oxidative modification, including disulfide bond formation and production of sulfenic, sulfinic, and sulfonic acids (30). DTT is capable of reversing disulfide bonds and sulfenic acid but not further oxidative states. Because RCS can lead to general protein misfolding, we reason that we are below this threshold, as the MIG reporter requires that GFP remain folded (33). We hypothesize that catalytic cysteines of *M. smegmatis* DnaBi1 and *M. leprae* DnaBi are the targets of RCS, leading to either disulfide bonding, as observed previously (25), or sulfenic acid production, both of which are reversible.

DnaB forms a hexamer and couples ATP hydrolysis to the unwinding of the double helix during replication. It is unclear whether *M. smegmatis* or *M. leprae* DnaB can hexamerize, bind to DNA, or hydrolyze ATP prior to splicing. The DnaB inteins examined in this study are located in the ATP-coordinating P-loop of the DnaB ATPase domain. For the archaeal recombinase/ATPase RadA from *Pyrococcus horikoshii*, which also houses an intein within the P-loop, ATPase activity was disrupted without splicing (18). Therefore, ATP hydrolysis, and thus replication, is unlikely without prior splicing. It is unclear, however, whether other activities, such as hexamer formation or DNA binding, are possible as a precursor. Interestingly, the RadA precursor from *P. horikoshii* retained the ability to bind ssDNA (22).

*M. smegmatis* DnaBi1 appears to sense multiple oxidative stressors. In the phagolysosome, mycobacteria encounter a variety of oxidants, including $H_2O_2$, HOCl, and chloramines such as NCT. Complicating matters, $H_2O_2$ is converted to HOCl with chloride by MPO, and chloramines result from HOCl reactions with amines (30). While chloramines and $H_2O_2$ are more specific in targeting cysteines, HOCl is more reactive. Regardless of whether ROS or RCS are most import *in vivo*, both are effective in blocking splicing, and both modifications can be reversed under reducing conditions. Following treatment with $H_2O_2$, *M. smegmatis* DnaBi1 forms an intramolecular disulfide bond between the catalytic C-118 and a noncatalytic cysteine (C-48) (25). Following treatment with NCT and HOCl, a similar patten is observed, suggesting that RCS may also lead to this intramolecular disulfide bond (25).

The *M. leprae* DnaB intein exhibits a different inhibition pattern upon RCS treatment, resulting in a HMW product (Fig. 3). A key difference between the *M. leprae* and *M. smegmatis* inteins is the number of noncatalytic cysteines. While the *M. smegmatis* intein has a single noncatalytic cysteine (C-48), the *M. leprae* intein has three (C-48, C-81, and C-110). These cysteines are not found within the conserved splicing blocks (Fig. 1C). The HMW product is consistent in size with a multimer of precursors. Importantly, the HMW product does not appear to represent nonspecific aggregation, as splicing is restored upon DTT treatment, the product remains soluble, and GFP maintains fluorescence. We speculate that this product is most likely a multimer formed by intermolecular disulfide bonds between cysteines on different precursors. It is curious that these extra cysteines are present in the *M. leprae* DnaB intein, given that they are not conserved with the *M. smegmatis* DnaB intein or predicted to be directly involved in the splicing mechanism.

Strategies to temporarily pause DNA replication under RCS stress could be beneficial when cells encounter RCS in nature. We propose that these DnaB inteins can sense RCS to pause replisome activity under this stress, to ensure chromosomal integrity. Once this assault has passed and conditions improve, RCS-induced oxidation can be reversed, allowing splicing to occur, which in turn permits immediate replisome activity. Another oxidant, the ROS $H_2O_2$, has also been shown to reversibly block *M. smegmatis* DnaBi1 protein splicing (25). Notably, ROS stress has been shown in human cells to slow the progression of the replication fork in order to protect the genome from damage (41). Indeed, while RCS react more readily with cysteines, they can modify nucleic acids and thus lead to genome damage (30). The ability to detect powerful

antimicrobials such as RCS and ROS and quickly pause any new replication could be important for persistence in harsh environments such as within the phagolysosome. Further, bacteria encounter stress from RCS such as HOCl and chloramines in a wide range of environments (30), where this strategy could also be beneficial. Therefore, in our model, DnaB splicing is blocked in the presence of RCS and ROS, allowing an immediate mechanism to halt new replisome formation and thus mitigate genome damage. Then, under more favorable reducing conditions, protein splicing and ultimately replication can resume. In summary, we have demonstrated that the activity of two mycobacterial DnaB inteins can be blocked by RCS, that this inhibition is fully reversible, and that *M. smegmatis* DnaBi1 splicing is responsive to RCS within *M. smegmatis* cells. These findings add to the rapidly expanding list of environmental conditions encountered by intein-containing bacteria that modulate protein splicing. Further, this work shows that mycobacterial DnaB inteins are capable of sensing a variety of stress conditions relevant to survival of the organism in nature (25, 27).

## MATERIALS AND METHODS

**Plasmids and strains.** Plasmids pACYC MIG DnaBi1, pACYC MIG *Mle* DnaBi, pMBC-283 Kan$^R$, and pMBC-283-Kan$^R$-DnaBi1WT were reported previously (25, 27). pACYC MIG DnaBi1, which expresses *M. smegmatis* DnaBi1 in the MIG reporter, and pACYC MIG *Mle* DnaBi, which expresses *M. leprae* DnaBi in the MIG reporter, were transformed into *Escherichia coli* BL21(DE3) cells for protein expression. pMBC-283 Kan$^R$ and pMBC-283 Kan$^R$-DnaBi1WT were transformed into *M. smegmatis* MC$^2$ 155 cells as described (34) for KISR survival assays described below.

**Protein expression.** *Escherichia coli* BL21(DE3) cells expressing either *M. smegmatis* DnaBi1 or *M. leprae* DnaBi in the MIG reporter were grown at 37°C in LB with 25 $\mu$g/ml chloramphenicol to the mid-logarithmic phase (optical density at 600 nm [OD$_{600}$] of ~0.5), and protein expression was induced by the addition 1 mM isopropyl-$\beta$-D-1-thiogalactopyranoside (GoldBio). Protein expression proceeded for 1 h at 37°C, cells were pelleted by centrifugation, and pellets were frozen at −20°C.

**Protein splicing.** Cells having previously expressed the MIG reporter were resuspended in 1× phosphate-buffered saline (PBS) (137 mM NaCl, 10 mM phosphate, 2.7 mM KCl [pH 7.4]) and lysed by sonication. Insoluble material was removed by centrifugation, and the soluble lysate was used to monitor splicing. Lysates were mixed with either double-distilled water (ddH$_2$O), HOCl, or NCT; following a 60-s incubation, ddH$_2$O or 50 mM DTT was added as indicated in the figure legends. Reaction mixtures were then incubated for ~20 h at 20°C to 22°C to permit splicing. Finally, products were measured using in-gel GFP fluorescence measurements, as described below.

**In-gel GFP fluorescence measurements.** Reaction mixtures were mixed with Laemmli sample buffer (Bio-Rad) to a final concentration of 1×. Samples were not heated, to maintain GFP structure, and products were separated using 8% to 16% Tris-glycine TGX gels (Bio-Rad). Gels were nonreducing unless indicated in the figure legends. Fluorescent products were detected immediately following SDS-PAGE, using an Amersham Imager 680 (GE Healthcare) (33).

**KISR survival assays.** *M. smegmatis* MC$^2$ 155 cells housing pMBC-283 Kan$^R$ or pMBC-283 Kan$^R$-DnaBi1WT (25, 27, 33) were grown to stationary phase (~3 days at 37°C) in Middlebrook 7H9 broth with albumin-dextrose-catalase (ADC) growth supplement, and cells were pelleted and resuspended in Middlebrook 7H9 broth lacking ADC growth supplement. Dilutions of OD$_{600}$ of $10^{-2}$, $10^{-3}$, and $10^{-4}$ were prepared in Middlebrook 7H9 broth lacking ADC growth supplement, and 100 $\mu$l of cells was plated and grown for 5 days at 37°C. Middlebrook 7H10 agar plates lacking oleic-albumin-dextrose-catalase (OADC) growth supplement were prepared fresh on the same day as plating and contained 50 $\mu$g hygromycin B (for plasmid maintenance), 150 $\mu$g/ml kanamycin sulfate, and, where indicated, 200 $\mu$M NCT.

## ACKNOWLEDGMENTS

We are grateful to Marlene Belfort and Michael J. Gray for useful discussions, as well as to Markus Nagl for the gift of NCT.

This work was supported by National Institutes of Health/KY INBRE grant P20GM103436, National Institutes of Health grant R15GM143662, and by start-up funds from Murray State University to C.W.L.

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
