## [Reviewer comments · Microbiology Spectrum]

**Microbiology
Spectrum**

Reactive chlorine species reversibly inhibit DnaB protein splicing in mycobacteria

Christopher Lennon, Daniel Wahl, J.R. Goetz, and Joel Weinberger II

Corresponding Author(s): Christopher Lennon, Murray State University

Review Timeline:

Submission Date:	May 17, 2021
Editorial Decision:	June 23, 2021
Revision Received:	August 16, 2021
Accepted:	August 26, 2021

Editor: Kathryn Elliott

Reviewer(s): Disclosure of reviewer identity is with reference to reviewer comments included in decision letter(s). The following individuals involved in review of your submission have agreed to reveal their identity: Jan-Ulrik Dahl (Reviewer #1)

Transaction Report:

DOI: <https://doi.org/10.1128/Spectrum.00301-21>

Prof. Christopher W Lennon
Murray State University
Biology
2111 Biology Building
Murray, KY 42071

Re: Spectrum00301-21 (Reactive chlorine species reversibly inhibit DnaB protein splicing in mycobacteria)

Dear Prof. Christopher W Lennon:

Thank you for submitting your manuscript to Microbiology Spectrum. Three expert referees have reviewed it, and in the text below you will find their comments. In general, the reviewers found your work on RCS impacts on protein splicing in mycobacteria to be an interesting topic that merits study. However, the reviewers raised some significant concerns. In particular, reviewer #3 questioned whether TRIS buffers were appropriate for studies with HOCl. Reviewer #2 also indicated that the manuscript would benefit from discussion of the potential mechanism of RCS toxicity as well as exploration of the high molecular weight species seen in the protein gels. Taken together, it would seem that addressing these comments would require more than a straightforward revision of the text. As a result, I regret to inform you that we will not be able to publish this manuscript in Spectrum.

I am sorry to convey a negative decision on this occasion, but I hope that the enclosed reviews are useful. We wish you well in publishing this report in another journal and hope that you will consider Spectrum in the future.

Sincerely,

Kathryn Elliott
Editor, Microbiology Spectrum

Reviewer comments:

Reviewer #1 (Comments for the Author):

This is short study describing the inhibitory effects of reactive chlorine species in protein splicing that I very much enjoyed reading.

My comments are the following:

- 1) Results, 3rd paragraph: "DTT is capable of ..." sentence is incomplete.
- 2) The reviewer is wondering whether the inhibition is specific to RCS? Given that HOCl-mediated

cysteine oxidation occurs at a high rate constant and hydrogen peroxide, another oxidant generated by innate immune cells, is much more slow acting (yet more thiol-specific), it would be intriguing to see whether the observation is a general phenomenon upon exposure to oxidative stress.

3) Fig. 4C: This is just a suggestion for future studies. The authors could perform the same experiment in liquid culture, which would allow them to measure kanamycin resistance more quantitatively (e.g. by quantifying the lag phase extension of each strain).

Reviewer #2 (Comments for the Author):

This manuscript describes a very straightforward set of experiments to show conditional protein splicing with the DnaBi inteins from *M. smegmatis* and *M. leprae*. The experiments are easy to follow, and the conclusions seem to be supported by the presented data. Although redox conditions have been shown to control cleaving in inteins before, this work is interesting in that it shows splicing control by known metabolites and relevant concentrations in a clinically meaningful context.

Despite these strengths, there are a couple of things what would strengthen the manuscript:

It would be good to explain the proposed mechanism of toxicity for RCS against the native host bacteria and hopefully clarify how pausing replication would be protective. It is implied that arresting cell division would protect the cells from RCS, but the central premise of the work could be better supported by more of a specific claim or mechanism.

How big is the high molecular weight species in Figure 3? Is it a simple multiple of the precursor, or is it likely to include cleaving or splicing products? Maybe it contains a branched intermediate? The methods section is also not completely clear on whether this was a reducing gel or a non-reducing gel, which makes it difficult to know if the HMW band comes from simple disulfide bonds or something more complicated. Adding a ladder to the gel would be helpful, or simply providing the observed molecular weights of the bands. Overall the manuscript could use some additional discussion of this species.

It is also interesting in Figure 3 that the cleaved GFP in the HOCl and NCT lanes also seems to shift to a higher apparent molecular weight, but this is not mentioned. There also appears to be a very faint HMW band appearing in the corresponding lanes of Figure 2. These results are very interesting, and some additional information at least on their size and potential origin would be interesting.

There is a sentence fragment, "Dithiothreitol (DTT) is capable of ," that appears to be left over from an earlier edit. This should be removed.

Reviewer #3 (Comments for the Author):

This is a potentially interesting paper looking at the effect of reactive chlorine species on protein splicing in mycobacteria. However, the authors should use phosphate buffers when treating cells with HOCl, because this oxidant reacts rapidly with TRIS to form chloramines. They should also do dose responses to determine whether physiologically relevant concentrations of these oxidants inhibit splicing.

Minor use Zn²⁺

We thank the three reviewers for the comments to improve this manuscript. Below are our responses (in red) to the reviewer comments (in blue). We also have substantially expanded our discussion section. All changes have been made using track changes for ease of review (we provide both the final version as well as the tracked version).

Reviewer comments:

Reviewer #1 (Comments for the Author):

This is short study describing the inhibitory effects of reactive chlorine species in protein splicing that I very much enjoyed reading.

We thank the reviewer for this very positive opinion on our work.

My comments are the following:

1) Results, 3rd paragraph: "DTT is capable of ..." sentence is incomplete.

We thank the reviewer for catching this oversight and have fixed this by completing the sentence as follows: "Dithiothreitol (DTT), a reducing agent, is capable of reversing some RCS modifications."

2) The reviewer is wondering whether the inhibition is specific to RCS? Given that HOCl-mediated cysteine oxidation occurs at a high rate constant and hydrogen peroxide, another oxidant generated by innate immune cells, is much more slow acting (yet more thiol-specific), it would be intriguing to see whether the observation is a general phenomenon upon exposure to oxidative stress.

We previously reported results demonstrating that hydrogen peroxide inhibits *M. smegmatis* DnaBi1 splicing (Kelley DS, Lennon CW, Li Z, Miller MR, Banavali NK, Li H, Belfort M. 2018. Mycobacterial DnaB helicase intein as oxidative stress sensor. Nat Comm. 9:4363.) and we refer to this within the introduction. Additionally, as stated in the introduction, abundant myeloperoxidase converts H₂O₂ (along with Cl⁻) within immune cells is converted into HOCl. Therefore, we believe the inhibition of protein splicing is likely a general response to oxidative stress and add additional discussion about this within the manuscript:

3) Fig. 4C: This is just a suggestion for future studies. The authors could perform the same experiment in liquid culture, which would allow them to measure kanamycin resistance more quantitatively (e.g. by quantifying the lag phase extension of each strain).

We thank the reviewer for this suggestion and will attempt to examine inhibition using this method in future studies.

Reviewer #2 (Comments for the Author):

This manuscript describes a very straightforward set of experiments to show conditional protein splicing with the DnaBi inteins from *M. smegmatis* and *M. leprae*. The experiments are easy to follow, and the conclusions seem to be supported by the presented data. Although redox conditions have been shown to control cleaving in inteins before, this work is interesting in that it shows splicing control by known metabolites and relevant concentrations in a clinically meaningful context.

We thank this reviewer for this positive feedback on our work.

Despite these strengths, there are a couple of things what would strengthen the manuscript: It would be good to explain the proposed mechanism of toxicity for RCS against the native host bacteria and hopefully clarify how pausing replication would be protective. It is implied that arresting cell division would protect the cells from RCS, but the central premise of the work could be better supported by more of a specific claim or mechanism.

We agree and have addressed this comment by substantially increasing analysis of our proposed model in the discussion section.

How big is the high molecular weight species in Figure 3? Is it a simple multiple of the precursor, or is it likely to include cleaving or splicing products? Maybe it contains a branched intermediate? The methods section is also not completely clear on whether this was a reducing gel or a non-reducing gel, which makes it difficult to know if the HMW band comes from simple disulfide bonds or something more complicated. Adding a ladder to the gel would be helpful, or simply providing the observed molecular weights of the bands. Overall the manuscript could use some additional discussion of this species.

To address the comment regarding, we have added MW markers to the gels. Please note that because we are measuring GFP fluorescence, the markers are not visible on the gels shown in figures.

Additionally, we have clarified our methods section to indicate that reducing agents are only present when indicated in the figure legends.

We thank the reviewer for making this point. In response, we now address the HMW product in detail within the discussion section and provide further analysis, concluding that the product is likely a multimer of intermolecular disulfide bonded precursors.

It is also interesting in Figure 3 that the cleaved GFP in the HOCl and NCT lanes also seems to shift to a higher apparent molecular weight, but this is not mentioned. There also appears to be a very faint HMW band appearing in the corresponding lanes of Figure 2. These results are very interesting, and some additional information at least on their size and potential origin would be interesting.

We thank the reviewer for pointing this out. We have added explanation of these products in the results and discussion section.

There is a sentence fragment, "Dithiothreitol (DTT) is capable of ," that appears to be left over from an earlier edit. This should be removed.

We thank the reviewer for pointing out this oversight. As described above under reviewer 1, we have completed this sentence.

Reviewer #3 (Comments for the Author):

This is a potentially interesting paper looking at the effect of reactive chlorine species on protein splicing in mycobacteria. However, the authors should use phosphate buffers when treating cells with HOCl, because this oxidant reacts rapidly with TRIS to form chloramines.

We thank the reviewer for this comment. In response, we have repeated all of our treatments HOCl and NCT in phosphate buffer (Fig. 2 and 3). As expected, the results are very similar to previous treatments, with inhibition by both RCS clear and reversible. Further, we discuss chlormamines and NCT in detail to avoid any confusion.

They should also do dose responses to determine whether physiologically relevant concentrations of these oxidants inhibit splicing.

We have expanded our experiments to include a dose response as suggested (Fig. 2 and 3). We observe complete inhibition at for both *M. smegmatis* and *M. leprae* DnaB inteins at 250 μ M HOCl and NCT.

As described within the introduction:

- “Within neutrophils, myeloperoxidase (MPO) produces millimolar quantities of HOCl from hydrogen peroxide (H₂O₂) and chloride.

As well as within the discussion:

- “MPO, which can represent 5% of the total protein in neutrophils, can produce levels of HOCl that can reach 25-50 mM.”

Therefore, the concentrations of RCS that inhibit *M. smegmatis* and *M. leprae* DnaB splicing are orders of magnitude below the physiological levels produced. We have also added to the discussion section to clarify this point.

Minor use Zn²⁺

We thank the reviewer for the comment and have corrected this mistake to ensure Zn²⁺ is consistently used.

August 26, 2021

Prof. Christopher W Lennon
Murray State University
Biology
2111 Biology Building
Murray, KY 42071

Re: Spectrum00301-21R1-A (Reactive chlorine species reversibly inhibit DnaB protein splicing in mycobacteria)

Dear Prof. Christopher W Lennon:

Thank you for thoroughly addressing the reviewer comments. Your manuscript has been accepted, and I am forwarding it to the ASM Journals Department for publication. You will be notified when your proofs are ready to be viewed.

Sincerely,

Kathryn Elliott
Editor, Microbiology Spectrum
